# Pullet Rearing Affects Collisions and Perch Use in Enriched Colony Cage Layer Housing

**DOI:** 10.3390/ani10081269

**Published:** 2020-07-25

**Authors:** Allison N. Pullin, S. Mieko Temple, Darin C. Bennett, Christina B. Rufener, Richard A. Blatchford, Maja M. Makagon

**Affiliations:** 1Center for Animal Welfare, Department of Animal Science, College of Agricultural and Environmental Sciences, University of California–Davis, 1 Shields Avenue, Davis, CA 95616, USA; apullin@ucdavis.edu (A.N.P.); cbrufener@ucdavis.edu (C.B.R.); rablatchford@ucdavis.edu (R.A.B.); 2Animal Behavior Graduate Group, College of Biological Sciences, University of California–Davis, 1 Shields Avenue, Davis, CA 95616, USA; 3Department of Animal Science, College of Agriculture, Food and Environmental Sciences, California Polytechnic State University, 1 Grand Avenue, San Luis Obispo, CA 93407, USA; smtemple@calpoly.edu (S.M.T.); dbenne06@calpoly.edu (D.C.B.)

**Keywords:** rearing, aviary, cage, pullet, laying hen, accelerometer, collision, perch, behavior

## Abstract

**Simple Summary:**

Early-life experiences for laying hens occur in the pullet rearing environment. Hens reared in aviaries use vertical space more than hens reared in non-enriched cages, but this effect has only been studied up to 23 weeks of age. Additionally, hens reared in aviaries sustain fewer keel bone fractures than those reared in non-enriched cages through the age of 73 weeks. Fractures are associated with hens having collisions with structures in their environment, but the long-term effect of rearing on collisions is not known. Lohmann LSL-Lite hens were reared in either aviaries or non-enriched cages until 19 weeks of age, then moved into enriched colony cages. Video recordings at 21, 35, and 49 weeks of age were used to identify behaviors associated with acceleration events for hens fitted with tri-axial accelerometers, as well as the proportion of birds utilizing elevated perches at two different heights. Our results indicate that hens reared in non-enriched cages experience more collisions than aviary-reared hens. Aviary-reared hens also prefer to utilize a higher perch than the cage-reared hens. These results suggest that rearing has long-term effects on space use and the ease with which hens transition among vertical spaces.

**Abstract:**

Hens reared in aviaries (AVI) as pullets have improved spatial abilities compared to hens reared in non-enriched cages (CON). However, this effect on behavior has been shown only to 23 weeks of age. Lohmann LSL-Lite hens were reared in either CON or AVI until 19 weeks of age and then moved into enriched colony cages (ECC) containing two elevated perches of different heights (*n* = 6 ECC/treatment). Focal hens (3 per ECC) were fitted with tri-axial accelerometers to record acceleration events at 21, 35, and 49 weeks of age. Video recordings from each age were used to identify behaviors associated with acceleration events as well as the proportion of hens utilizing perches. CON hens experienced more acceleration events (*p* = 0.008) and more collisions (*p* = 0.04) than AVI hens during the day at 21 and 35 weeks of age. The total proportion of hens perching at night was similar between treatments across most time points, but fewer CON hens used the high perch compared to AVI hens throughout the study (*p* = < 0.001). Rearing in aviaries influences hen behavior out to peak lay for collisions and out to mid-lay for perch height preference in ECC.

## 1. Introduction

The pullet rearing environment, which constitutes the first 16–18 weeks of life, can have long-term effects on laying hen welfare [1]. Studies of pullet housing have shown that the rearing environment affects cognition, behavior, as well as bone and muscle development. For example, compared to hens reared in non-enriched cages, aviary-reared hens show improved working memory in the reversal phase of a holeboard task [2]. Birds reared with access to vertical space use perches and elevated platforms more at 19 weeks of age than birds reared without elevated structures [3,4]. Aviary-reared pullets have stronger, more developed radius, humerus, and tibia bones, a larger keel bone, and heavier wing and breast muscles compared to pullets from non-enriched cages at 16 weeks of age [5]. Importantly, these effects on the skeletal system persist through to the end of the laying period. Aviary-reared hens have stronger bones with greater cross-sectional areas and higher mineral content compared to hens reared in non-enriched cages at 73 weeks of age [6].

Rearing environments have also been shown to affect keel bone fracture development. Hens housed in enriched colony cages had a higher incidence of keel bone fractures at the end of the laying period if they were reared in non-enriched cages versus aviaries [7]. The keel bone is crucial for flight and general mobility as it serves as an attachment site for wing muscles [8]. Birds with fractured keels may experience pain [9], reduced mobility [9,10], and reduced egg production [11,12]. Keel bone fractures are suspected to be partly attributed to physical trauma to the bone [13,14]. Accordingly, previous research found the number of collisions that a hen experienced corresponded with the risk of keel fracture development [15,16]. Using tri-axial accelerometers, the potential causes of such trauma were examined [17]. The keel-mounted sensors registered acceleration events experienced at the keel by hens as they navigated their enriched colony cage environments. Collisions with cage structures, and specifically the perch, caused most (~80%) of the acceleration events. However, the impact of the rearing environment on collisions was not explored.

The reported influence of rearing on keel bone fracture prevalence [7] may reflect differences in the birds’ abilities to navigate their adult environments. Therefore, the objective of this study was to investigate whether the early rearing environment (non-enriched cage vs. aviary housing) affects the behavior of hens after they are placed in their layer environment. Specifically, we aimed to determine whether pullet rearing affects the incidence of acceleration events recorded at the keel, and the behaviors associated with these events. We further explored actions and housing structures associated with observed collisions. Considering that the perch was the main structure associated with collisions in a previous study [17], we also investigated the effect of rearing on perching behavior focusing on perch use and the distribution of birds across the two available perches.

## 2. Materials and Methods

### 2.1. Ethical Statement

All procedures were approved by the California Polytechnic State University Animal Care and Use Committee (Protocol #1613).

### 2.2. Animals

On farm data collection was conducted at the Cal Poly Poultry Center at California Polytechnic State University. A total of 301 pullets were used in this study. The pullets were selected from among 3000 day-old Lohmann LSL-Lite chicks, with infra-red beak trim, that were obtained from Hy-Line North America, LLC (Burley, ID, USA). All pullets were from the same parent flock.

### 2.3. Pullet Housing and Management

A total of 2000 pullets were reared in a conventional non-enriched cage system (CON; 98 × 69.5 × 41 cm, L × W × H; SALMET S1000 Rearing System, SALMET GmbH & Co. KG, Dietzenbach, Germany). The wire floored cages were stacked in three tiers of 36 cages/tier. All pullets were initially housed in the middle tier at 55–60 pullets/cage with a space allowance of 113.5–123.8 cm^2^/pullet. Space per bird was increased during wk 6, at which time pullets were sorted among cages in the top and middle tiers. The pullets were redistributed among all three tiers at 12 weeks of age, with 20 pullets/cage at 340.6 cm^2^/pullet. Water was provided ad libitum via automatic water lines (Ziggity, Middlebury, IN, USA) with 4 nipple drinkers/cage. An automated trough feeder provided hens with ad libitum access to a commercial diet.

A total of 1000 pullets were reared in a multi-tier aviary system (AVI; 16.5 × 4.7 × 2.5 m, L × W × H; SALMET Pedigrow 2 Rearing System, SALMET GmbH & Co. KG, Dietzenbach, Germany). The system contained three elevated tiers at 0.3 m, 1.2 m, and 1.6 m height from the floor. Pullets were enclosed into the middle tier for the first three weeks (564.9 cm^2^/pullet). After the first three weeks, tier doors were opened and pullets could access the bottom and top tiers of the system, shelves on the outer edge of the middle and top tiers (147 × 18 cm/shelf, L × W), and a scratch area adjacent to the system containing pine wood shavings over a concrete floor (514,000 cm^2^ scratch area). Considering all sections, shelves, and adjacent scratch area, the total available floor space was 2267 cm^2^/pullet for the remainder of the rearing period.

Each tier in AVI contained a wire mesh floor, manure belt, automatic water line, automatic feeder line, and square metal perch (2.5 cm diameter) above the feeder. The top tier contained an additional 11 square metal perches (27.7 cm/pullet of linear perch space). Wire mesh ramps were provided to facilitate access from the floor to the middle tier. Water was provided ad libitum with 132 nipple drinkers/tier (Ziggity). AVI chicks were fed the same commercial diet as CON chicks.

For both rearing environments, temperature and artificial lighting were maintained according to the North American Edition of the LSL-Lite Layers Management Guide (Lohmann Tierzucht GmbH, Cuxhaven, Germany).

### 2.4. Layer Housing and Management

At 19 weeks of age, all birds were moved into their adult housing system—a two-story barn containing 128 enriched colony cages (ECC; 64 cages/floor; 25 ± 2 hens/ECC). Birds were placed in ECC with other birds from the same rearing treatment, and treatments were distributed randomly across each story of the barn. A total of 12 ECC were chosen as focal cages (*n* = 6 ECC/rearing treatment). Each ECC (360 × 63 × 56 cm, L × W × H; SALMET AGK 3600 Enrichable Colony System, SALMET GmbH & Co.) contained two elevated round metal perches (9 and 25 cm from cage floor) that were each 225 cm in length, as well as an enclosed nest area (51 × 30 × 52 cm, L × W × H) and scratch pad (35 × 13 cm, L × W) over a wire mesh floor. The hens had access to 907.2 cm^2^/hen of floor space and 18 cm/hen of perch space. Water was provided ad libitum with 8 nipple drinkers/cage (Ziggity). A commercial diet was provided ad libitum in automatic feeders which ran along each cage row. The artificial lighting program started at 12 h/d at initial placement and was subsequently stepped up by 1 h/week to reach 16 h/day.

### 2.5. Behaviors Associated with Acceleration Events at the Keel

At 21, 35, and 49 weeks, three randomly selected focal hens/ECC (36 total hens/age) were fitted with a custom jacket containing a tri-axial accelerometer with a sensor positioned over the keel bone (Custom Idea Ltd., Shepton Mallet, UK; for more information on jackets and accelerometers, see [17]). A different set of focal hens was selected at each age. Jackets were fitted to hens during nighttime hours in order to minimize hen disturbance. Each white jacket contained a unique black symbol, which allowed us to individually identify the focal hens during recording. Jackets and accelerometers were worn continuously for 11 d at each age. Hens habituated to the jackets during the first day, and data were recorded for the remaining 10 d. Accelerometers were programmed to record events at the keel with a combined magnitude of acceleration >12 G-Units (G) to avoid the inclusion of minor body movements [17]. The combined magnitude of acceleration was calculated as the square root of the sum of squared acceleration in three dimensions (X, Y, and Z) [17]. At the conclusion of each age trial, a log containing date, time, and the maximum combined magnitude of acceleration for each event was generated for each hen. This log was used for video analysis.

ECC were video recorded continuously during each of the three 10 d trials using two video cameras per cage (Veilux 700 Line Day Night IR Bullet Camera, Veilux Inc., Grand Prairie, TX, USA) connected to a digital video recorder with surveillance software (GeoVision Digital Surveillance System version 8.5, GeoVision Inc., Taipei, Taiwan). After the end of each age trial, trained observers used the accelerometer log with the video recordings to identify the behavior associated with each acceleration event (Table 1) [17]. Observers were blind to rearing treatment, randomly assigned to video from both rearing treatments, and analyzed all videos using GeoVision Software (GeoVision Digital Surveillance System, ViewLogEx v.8.5.6.0, GeoVision Inc., Tapei, Taiwan). Inter-observer reliability was high (≥90% inter-observer agreement between trainer and each trainee of approximately 20 acceleration events).

### 2.6. Perching Behavior

Hen perching behavior was evaluated from video on the second and sixth day of each 10-d trial. Using instantaneous scan sampling with 5-min intervals, the number of hens using the low and high perch were observed. It was previously reported that 5-min interval sampling was representative of perching behavior throughout a 15 h day [18]. A total of five hours of video was reviewed on each day: three hours while the house lights were on (06:00–06:55, 12:00–12:55, 16:00–16:55; “day”) and two hours while lights were off (23:00–23:55, 02:00–02:55; “night”). Preliminary analysis (a review of 24 h of video data for five ECC) found this sampling scheme to be representative of the entire day for perching behavior.

### 2.7. Data Analysis

For all data analyses, R statistical software (version 3.5.2, R Core Team, Vienna, Austria) [19] and linear mixed effect models (lme4 package) [20] were used. Model assumptions were checked for normality of residuals by visually evaluating Q–Q plots, residual vs. fitted values plots, and homoscedasticity of independent variables. As models did not meet one or more normality criteria, generalized linear mixed effect models with a Poisson distribution (lme4 package) [20] were used to analyze the total number of acceleration events and total number of collisions sustained by each hen across the 10 d at each age. Model assumptions for generalized linear mixed effects models were checked using package DHARMa [21]. Following a previous study [17], only acceleration events and collisions associated with accelerations >20 G were included in the statistical analyses. The effect of treatment (factor with 2 levels: CON, AVI), age (factor with 3 levels: 21–22, 35–36, and 49–50 weeks), day/night (factor with 2 levels: day, night), and their interactions (treatment × age, treatment × day/night, age × day/night, treatment × age × day/night) were assessed, with hen ID nested in age nested in ECC as a random effect. The final models were obtained by a stepwise backwards reduction using ANOVA for model comparison with a *p*-value of >0.05 as the criterion of exclusion.

Within each 5-min observation, we calculated the average proportion of hens from each ECC that utilized either of the two perches. The numerator was the total number of hens on the low perch and high perch, and the denominator was the total number of hens in the ECC. The data were analyzed using a zero-inflated beta regression model (glmmTMB package) [22]. The effect of treatment (factor with 2 levels: CON, AVI), age (factor with 3 levels: 21–22, 35–36, and 49–50 weeks), day/night (factor with 2 levels: day, night), and their interactions (treatment × age, treatment × day/night, age × day/night, treatment × age × day/night) were assessed, with day nested in age nested in cage as a random effect.

We also calculated the average proportion of perching hens from each ECC that utilized the high perch during nighttime hours. The numerator was the total number of hens on the high perch, and the denominator was the total number of hens on the low perch and high perch. The same zero-inflated beta regression model described previously was used, excluding the effect of light.

## 3. Results

### 3.1. All Acceleration Events: Descriptive Summary

At 21–22 weeks of age, we obtained accelerometer data for 31 hens (16 CON hens across 6 ECC; 15 AVI hens across 6 ECC). Data from 5 of the 36 focal hens were excluded as their accelerometers failed to record or store data properly. A total of 470 acceleration events were recorded during this 10-d data collection period. Of the 470 recorded acceleration events, 393 (83.6%) were successfully matched to video. The remaining 16.4% fell into the Cannot ID category (Table 1) and were excluded from further analysis. A previous study using the same accelerometers identified acceleration events 12–20 G to be associated mainly with general hen movements (e.g., 45.8% grooming) [17]. Similarly, in our study, acceleration events of 12–20 G were mainly general movement events (Figure 1). Therefore, statistical analysis focused on the 209 acceleration events >20 G, which represented 16 CON and 13 AVI hens from the 12 ECC. Figure 1 shows the behavioral categories associated with these events. Collisions were the most frequent cause of acceleration events >20 G (59.3%), followed by aggression (19.1%), mass-scattering (12.4%), grooming (6.2%), and wing-flapping (3.0%).

For 35–36 weeks of age, a total of 267 acceleration events were recorded from 28 hens (14 CON hens across 6 ECC; 14 AVI hens across 6 ECC; 8 accelerometers failed to record properly). Of the 211 acceleration events that were matched to video, 70 had accelerations >20 G. Six of the hens only experienced collisions with accelerations 12–20 G. As a result, the analyzed data represented 13 CON and 9 AVI hens (from all 12 replicate ECC). Collisions were the most frequent cause of acceleration events >20 G (44.3%), followed by aggression (18.6%), grooming (18.6%), wing-flapping (14.3%), and mass-scattering (4.2%).

Lastly, for 49–50 weeks of age, a total of 172 acceleration events were recorded from a total of 26 hens over the 10-d data collection period (14 CON hens across 6 ECC; 12 AVI hens across 6 ECC). Ten of the initial 36 accelerometers failed during the course of data collection and their data were not included in analysis. Of the 153 identifiable acceleration events, 52 acceleration events had accelerations >20 G from 11 CON (6 ECC) and 8 AVI hens (5 ECC). Aggression was the most frequent cause of acceleration events >20 G (40.4%), followed by grooming (32.7%), and collisions (26.9%). Mass-scattering and wing-flapping were not associated with any acceleration events >20 G for this timepoint.

### 3.2. Incidence of Acceleration Events >20 G

Rearing treatment, age, and day/night had a three-way interactive effect on the incidence of acceleration events >20 G (χ^2^ = 9.59, df = 2, *p* = 0.008; Figure 2). During the day, both rearing treatments sustained the most acceleration events at 21 weeks of age. However, we observed a reduction in the number of acceleration events sustained by AVI hens at 35 weeks of age that remained fairly constant to 49 weeks of age. For CON hens, we observed that the number of acceleration events sustained decreased between 21, 35, and 49 weeks of age. Daytime observations also revealed that CON hens sustained more acceleration events than aviary-reared hens at 21 and 35 weeks of age (estimated means [95% CI]: 4.3 [2.8, 6.7], 2.8 [1.7, 4.7], 1.2 [0.6, 2.1] acceleration events/hen/10 day in CON birds vs. 3.4 [2.1, 5.5], 0.7 [0.3, 1.4], 1.3 [0.7, 2.6] acceleration events/hen/10 d in AVI birds at 21, 35 and 49 weeks of age, respectively). At night, AVI hens had a consistently low number of acceleration events sustained across all ages (0.6 [0.3, 1.3], 0.2 [0.08, 0.7], 0.1 [0.04, 0.5] acceleration events/hen/10 d at 21, 35, and 49 weeks of age). Moreover, CON hens sustained the greatest number of nighttime acceleration events at 21 weeks of age, and then the acceleration events decreased and remained low at 35 and 49 weeks of age (1.8 [1.1, 2.9], 0.2 [0.05, 0.6], 0.06 [0.008, 0.4] acceleration events/hen/10 d at 21, 35, and 49 weeks of age, respectively).

### 3.3. Incidence of Collisions >20 G

Rearing treatment, age, and day/night had a three-way interactive effect on the incidence of collisions that were associated with acceleration events >20 G (χ^2^ = 6.34, df = 2, *p* = 0.04; Figure 3). The patterns of the interaction were similar to the patterns reported for all acceleration events. During the day, AVI and CON hens experienced the most collisions at 21 weeks of age. However, the number of collisions experienced by AVI hens decreased by 35 weeks of age and stayed consistent at 49 weeks of age. In CON hens, the number of collisions sustained over time decreased across subsequent time points, and CON hens experienced more collisions than AVI hens at 21 and 35 weeks of age (estimated means [95% CI]: 2.0 [1.1, 3.6], 1.0 [0.5, 2.1], 0.2 [0.08, 0.7] collisions/hen/10 d in CON birds vs. 1.3 [0.7, 2.6], 0.2 [0.08, 0.7], 0.3 [0.09, 0.8] collisions/hen/10 d in AVI birds at 21, 35, and 49 weeks of age, respectively). During the night, AVI hens experienced minimal collisions across all ages (0.4 [0.2, 0.9], 0.1 [0.02, 0.4], 0.2 [0.05, 0.6] collisions/hen/10 d at 21, 35, and 49 weeks of age). CON hens experienced the most collisions during lights off at 21 weeks of age, but collisions decreased by 35 weeks of age and remained low when assessed at the end of the trial (1.2 [0.6, 2.3], 0.04 [0.006, 0.3], 0.04 [0.006, 0.4] collisions/hen/10 d at 21, 35, and 49 weeks of age).

### 3.4. Collision Objects and Actions: Descriptive Summary

Over the course of the study, a total of 117 collisions >20 G were recorded for CON hens across all cages and all time points. The majority of these were with the high (34.2%) and low perch (29.1%), followed by the floor (20.5%), other birds (14.5%), and the cage-wall (1.7%). CON hens experienced collisions by running into the object or conspecific (31.6%), slipping on a perch (27.4%), being pushed by another conspecific (23.1%), and ascending onto (15.4%) and descending from (2.5%) perches. For AVI hens, a total of 52 collisions with accelerations >20 G were recorded across all cages and all time points. The highest proportion of collisions were with the high perch (40.4%), followed by the floor (26.9%), the low perch (17.3%), other birds (11.5%), and the cage-wall (3.9%). The actions associated with collisions for AVI hens were slipping on a perch (34.6%), ascending to perches (23.1%), being pushed by another conspecific (21.2%), running into the object or conspecific (17.3%), and descending from perches (3.8%).

### 3.5. Perching Behavior

Rearing, age, and day/night had a three-way interactive effect on perch use (χ^2^ = 8.57, df = 2, *p* = 0.01; Figure 4). As indicated in Figure 4, daytime perch use was low in both treatments. At night, the proportion of birds perching was comparable across treatments at 21 and 35 weeks of age. At 49 weeks of age, a higher proportion of AVI hens were observed perching as compared to CON hens (0.317 [0.249, 0.393], 0.322 [0.254, 0.400], 0.352 [0.280, 0.432] proportion of CON hens perching/ECC/5-min observation vs. 0.290 [0.226, 0.363], 0.331 [0.261, 0.409], 0.489 [0.407, 0.572] proportion of AVI hens perching/ECC/5-min observation at 21, 35, and 49 weeks, respectively; estimated means [95% CI]).

Of the hens that were using perches, treatment and age affected the proportion of hens using the high perch specifically. More perching AVI hens utilized the high perch than perching CON hens throughout lay (χ^2^ = 11.1, df = 1, *p* = < 0.001; Figure 5). Regardless of rearing treatment, all hens increased use of the high perch at 49 weeks of age compared to 21 and 35 weeks (χ^2^ = 6.40, df = 2, *p* = 0.04).

## 4. Discussion

The objective of this study was to determine whether pullet rearing conditions affect the incidence of acceleration events and collisions at the keel, as well as perching behavior, for ECC housed laying hens. We predicted that AVI hens would sustain fewer acceleration events and fewer collisions than CON hens, and we predicted that AVI hens would perch more overall, and specifically on the high perch, than CON hens. We found that both AVI and CON hens experienced fewer acceleration events at the keel and collided with features of their environment less frequently as they aged. However, AVI hens sustained fewer acceleration events and fewer collisions than CON hens at initial and peak phase of the laying period (21 and 35 weeks of age), suggesting that AVI hens were navigating their laying environment with more ease from an earlier age than CON hens.

Previous research has demonstrated that rearing environments contribute to learning ability, such that birds reared in enriched environments learn faster than birds reared in non-enriched environments. Floor-reared pullets with one week of exposure to outdoor range were faster to learn a Y-maze task compared to floor-reared pullets with no other structural or sensory enrichment [23]. Similarly, floor-reared pullets with three weeks of sensory enrichment were faster to learn a T-maze task than floor-reared pullets without enrichment [24]. During the reversal phase of a holeboard task, pullets reared in a multi-tier aviary for 12 weeks were faster to complete the task than pullets reared in a conventional non-enriched cage for 16 weeks [2]. These previous findings suggest that enriched (AVI) rearing environments improve birds’ abilities to learn compared to non-enriched (CON) rearing environments, which may have contributed to AVI hens experiencing fewer acceleration events and collisions than CON hens at earlier time points. Cognitive testing was not performed in our experiment, but our findings indicate that more cognitive tests and potential brain analyses are needed to assess the effect of rearing on long-term learning in novel environments.

It is worth noting that the magnitude of difference between rearing treatments’ estimated means was low, with CON hens experiencing about 1–2 acceleration events/hen/10 day and about 1 collision/hen/10 day more than AVI hens at initial and peak lay. However, assuming that these numbers remained consistent in the 14 weeks between initial and peak lay, CON hens could have cumulatively experienced up to 10 collisions more per hen than AVI hens between 21 to 35 weeks of age. Previous work found that birds with a higher number of collisions were at an increased risk for sustaining keel bone fractures [15,16], and other researchers reported that CON hens have a higher incidence of keel bone fractures than AVI hens at 73 weeks of age in ECC [7]. Therefore, considering our data with these previous two studies, it is possible that CON hens cumulatively experience more collisions and thus are at a greater risk for injury. Future research should evaluate the effect of rearing on both collisions and keel bone fractures concurrently for further evaluation. As mentioned, the number of collisions experienced across the 10-d trials could be considered low, with estimated means ranging from 0–2 collisions/hen/10 day >20 G across rearing treatments and ages during the day. Our study found fewer collisions on average than previous work using the same accelerometers, which evaluated the number of collisions per hen across two 3-week timepoints/ECC [17]. Baker et al. [17] reported an average of 12.0 and 12.6 collisions/hen/3 weeks >20 G during the day (estimated 5.7 and 6.0 collisions/hen/10 d). The hens used in this previous work were floor-reared, older, of a different strain (Hy-line W36), and housed in ECC with a different spatial layout of perches than those in our study. Some or all of these factors could have contributed to the discrepancy in the number of collisions reported in our study and the previous one [17]. Overall, it is important to consider that acceleration events at the keel are experienced by individuals. The spread of the interquartile range of acceleration event data, particularly at 35 weeks of age (Figure 2), suggests that some CON individuals experienced more acceleration events than the majority of AVI hens, which could further increase their risk for keel bone fractures during peak lay.

Collisions constituted the main cause of acceleration events experienced at the keel bone for initial and peak lay. This finding confirms previous results obtained using the same accelerometers that collisions constitute a majority of acceleration events >20 G [17]. However, the number of collisions remained consistent over two age groups (52–60 and 74–83 weeks) in the previous results [17], whereas the collisions in our study decreased as birds aged. Other researchers have also found hens to experience fewer collisions with increasing age [14,25]. The inverse relationship between collisions and age has been attributed to hens being the most active earlier in the laying period and reducing activity over time [25,26,27,28]. An alternative explanation for our study could be that hens from both rearing treatments were experiencing a novel environment at initial lay. Neither of the rearing environments matched the laying environment in terms of group size, space allowance, and the design and spatial layout of physical structures. A mismatch between the rearing and laying environment is attributed to behavioral challenges when transitioning into the laying environment (e.g., aggression, space perception) [1]. As a result, having the most acceleration events and collisions at the initial phase of the laying period could also be attributed to adapting to a novel environment.

We descriptively evaluated the objects that birds collided with, as well as the actions that birds performed at the time of collision, to provide further detail into how birds utilized their space. Collisions with either of the perches constituted the greatest proportion of collision objects for both treatments (63.3% of collisions with low perch and high perch for CON hens vs. 57.7% of collisions with low perch and high perch for AVI hens). These findings are in agreement with previous research [17], which also found perches to be the main objects that birds collided with in ECC (71.5% of collisions with perches). The authors noted that collisions with perches occurred mainly as hens ascended onto them [17]. Along with slipping while perching, ascending onto perch was also among the top two actions associated with perch collisions by AVI hens in this study. Meanwhile, the top two collision action categories for CON hens were running into an object or a conspecific while locomoting across the floor and slipping on a perch.

We predicted that more AVI hens would be observed perching than CON hens at all ages. If true, this finding would explain why perch ascent may have been a leading source of collisions for AVI hens but not CON hens. Previous research found that pullets reared with access to vertical space utilized perches and elevated platforms more than birds reared without perches [3,4]. However, in our study, the total proportion of birds perching was similar across rearing treatments and for most time points. Although CON hens were not raised with perches, they may have perched on the water lines as pullets, offering one possible explanation for the similar perch use, including during the earliest observation time point. It is also possible that the two weeks that the birds spent in the ECC system before the data collection at 21 weeks of age provided the chicks with sufficient time to learn to use the perches. This is particularly likely since the low perch was located only 9 cm from the cage floor and ran the length of the cage, making it easily accessible for the birds.

Daytime perch use was low, with only 1–8% of birds observed perching, regardless of perch height or rearing treatment. A previous study reported a range of 17–30% of birds perching during lights on in groups of 33–50 hens per ECC [29]. Others reported birds spending 7–25% of their time perching during the day in groups of 14 hens per ECC [30]. In these studies, daytime perch use was attributed to short bouts of grooming, standing, sleeping, and facilitating access to another resource (e.g., water line). In our study, the perches did not facilitate access to feed or water, so it is possible that they were utilized less during the daytime while birds were active and using other resources in the ECC. On the other hand, laying hens are highly motivated to access perches at night for roosting, which is attributed to an antipredator defense strategy [31]. At least one-third of birds, regardless of rearing treatment, were observed roosting on perches at night at all three ages in our study. The remaining proportion of birds were commonly observed huddled together in one or two large groups on the floor of the ECC. The proportion of roosting birds in our study is lower than reported in several previous studies of nighttime perching among hens housed in enriched cages (53–72% of hens roosting [29]; 85–100% of hens roosting [32]). However, other researchers have reported that the majority of hens in ECC sleep huddled on the floor during lights off as opposed to roosting on perches [33]. Birds in our study had sufficient perch space for all hens to be perching simultaneously on either perch (at least 15 cm per hen [32]), but possibly different designs and layout of perches in ECC contributed to the behavioral differences in our study compared to others.

We evaluated perching birds’ preference for the low versus high perch. Given the low daytime perch use observed, our analysis focused on nighttime observations. While both AVI and CON hens utilized perches at night, more AVI hens used the high perch. Previous studies that evaluated the effect of rearing on perch and platform use collected data only at 16 and 19 weeks of age [3,4]. Our work demonstrates that rearing influences height preference into mid-lay (49 weeks of age). It is not clear from our study whether the influence of rearing on roost height preference reflects differences in spatial cognition, perception, learning, or physical abilities. It is also unclear whether it was the height or other aspects of perch design that were preferred by the AVI birds, or that may have dissuaded CON birds from mounting it. Unlike the low perch, the high perch in this study was suspended from the cage ceiling and consequently could sway back and forth. This instability of the perch could have impacted the birds’ preference for it.

In addition to perch use, the lighting period, whether lights were on or off, also had an effect on collisions. All birds, regardless of rearing treatment, experienced more collisions during the day versus during the night. Similarly, previous studies found that the majority of collisions took place during lights on, and specifically during the dusk and dawn phases when hens were moving to and from roosting [17,25]. In our study, we did not observe a pattern of collisions during dawn and dusk specifically. Instead, the higher prevalence of collisions during lights on could be attributed to hens being generally more active during the day than during the night. In laying hens, more illuminance promotes more physical activity [34], and, as a result, hens have more opportunities for collisions during lights on. Furthermore, the lights on period was twice as long as the lights off period in our study (16 h vs. 8 h, respectively), so hens also had more total time to sustain collisions during the lights on period as opposed to lights off.

Lastly, we found social interactions to be an important contributing factor to keel acceleration events, such that aggression was the second-most cause of acceleration events in another study and in our study [17]. Coupled with the fact that a notable collision object category was conspecifics and a notable collision action category was birds being pushed by conspecifics, these findings warrant further investigation into the relationship of social behavior and collisions. In the context of early experiences, it is important to consider that rearing environments differ in the group sizes allowance, availability of space for hiding or retreat, and other aspects that influence social experiences and development of chicks. Research into the interactions between rearing, social behavior, space use, and collisions is needed.

## 5. Conclusions

We proposed that the reported influence of rearing on keel bone fracture prevalence may be due to differences in birds’ abilities to move through their adult environment. Our study found that rearing environment does affect the number of acceleration events experienced at the keel, collisions associated with these acceleration events, and perch use. Collisions have been linked to keel bone fractures in prior research, and these behavioral effects seem to persist through peak lay for acceleration events and collisions and through mid-lay for perching. Keel bone fracture assessment was outside the scope of this study. However, higher keel bone fracture incidences were found during peak lay for ECC housed hens that had been reared in conventional non-enriched cages vs. aviaries [7]. It is possible that experiencing more collisions through peak lay, as demonstrated in our study, may explain these results. Future studies should evaluate collisions and keel bone fractures from the same individual hens to evaluate the link between rearing environment, collisions, and keel bone fractures.

## Figures and Tables

**Figure 1 animals-10-01269-f001:**
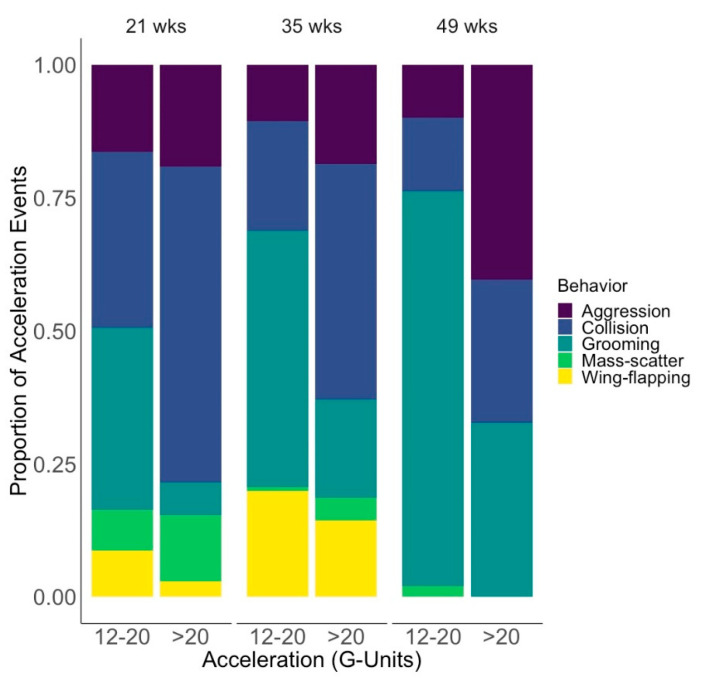
All identifiable acceleration events sustained by all hens at each age, grouped by behavioral category and acceleration category. At 21 weeks, the 12–20 G acceleration category is based on 184 events and the >20 G category is based on 209 events. At 35 weeks, 12–20 G represents 141 events while >20 G represents 70 events. Lastly, at 49 weeks, 12–20 G contains 101 events and >20 G contains 52 events.

**Figure 2 animals-10-01269-f002:**
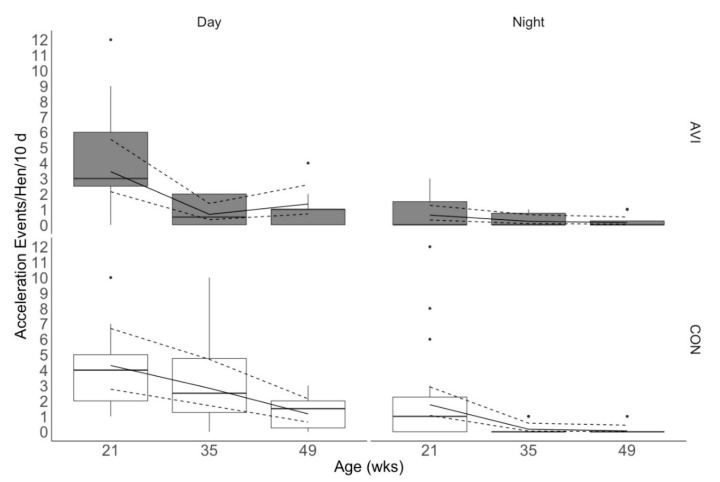
Total acceleration events >20 G sustained per hen over 10 d at each age during day (16 h/d) and night (8 h/day in enriched colony cages (ECC). Hens were reared in either conventional non-enriched cages (CON) or a multi-tiered aviary (AVI) until 19 weeks of age before being moved into ECC. Acceleration events consisted of collision, aggression, grooming, wing-flapping, and mass scattering events. Boxes represent the interquartile range, where the black line represents the median, the top of the box represents the 75th quartile, and the bottom of the box represents the 25th quartile. Whiskers represent the minimum and maximum values, while black dots indicate outliers. The solid line overlaying the box plots is the estimated mean values from the generalized linear mixed effects model, and the dashed lines are the 95% confidence intervals of the estimated means.

**Figure 3 animals-10-01269-f003:**
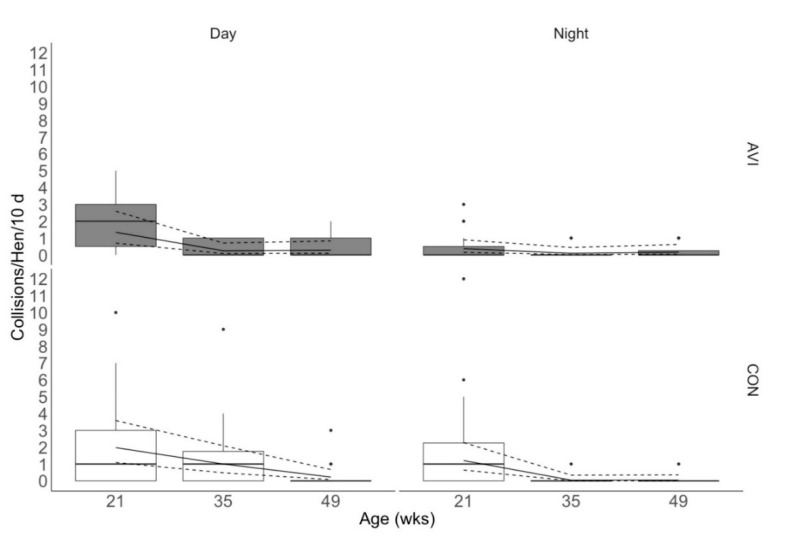
Total collisions >20 G sustained per hen over 10 d at each age during lights on (16 h/d) and lights off (8 h/day) in enriched colony cages (ECC). Hens were reared in either conventional non-enriched cages (CON) or a multi-tiered aviary (AVI) until 19 weeks of age before being moved into ECC. Collisions were with structural objects or conspecifics. Boxes represent the interquartile range, where the black line represents the median, the top of the box represents the 75th quartile, and the bottom of the box represents the 25th quartile. Whiskers represent the minimum and maximum values, while black dots indicate outliers. The solid line overlaying the box plots is the estimated mean values from the generalized linear mixed effects model, and the dashed lines represent the 95% confidence intervals of the estimated means. wks: weeks.

**Figure 4 animals-10-01269-f004:**
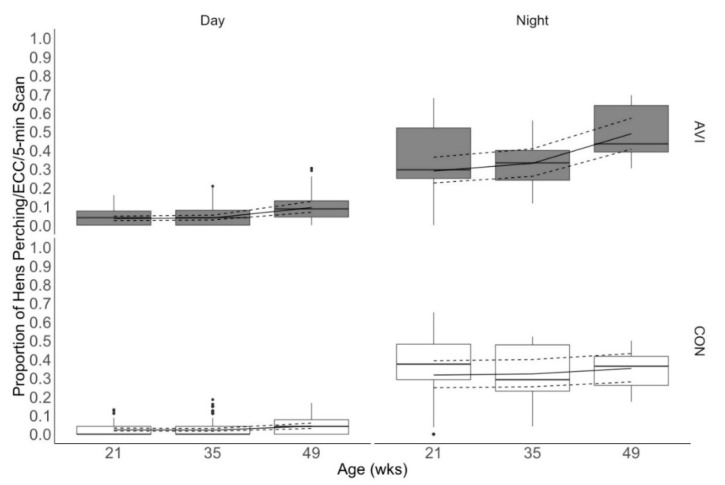
The proportion of birds perching per enriched colony cage at each age. Hens were reared in either conventional non-enriched cages (CON) or a multi-tiered aviary (AVI) until 19 weeks of age before being moved into enriched colony cages. Boxes represent the interquartile range, where the black line represents the median, the top of the box represents the 75th quartile, and the bottom of the box represents the 25th quartile. Whiskers represent the minimum and maximum values, while black dots indicate outliers. The solid line overlaying the box plots is the estimated mean values from the generalized linear mixed effects model, and the dashed lines are the 95% confidence intervals of the estimated means. wks: weeks.

**Figure 5 animals-10-01269-f005:**
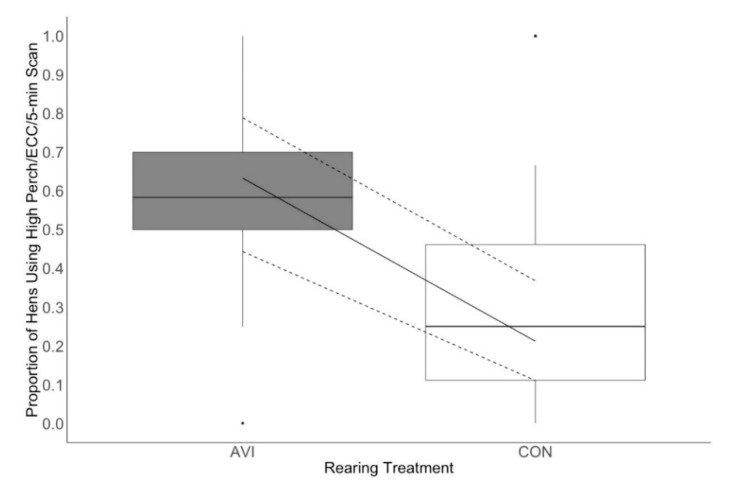
The proportion of perching birds on the high perch by rearing treatment throughout lay (21–49 weeks). Hens were reared in either conventional non-enriched cages (CON) or a multi-tiered aviary (AVI) until 19 weeks of age before being moved into enriched colony cages. Boxes represent the interquartile range, where the black line represents the median, the top of the box represents the 75th quartile, and the bottom of the box represents the 25th quartile. Whiskers represent the minimum and maximum values, while black dots indicate outliers. The solid line overlaying the box plots is the estimated mean values from the generalized linear mixed effects model, and the dashed lines are the 95% confidence intervals of the estimated means.

**Table 1 animals-10-01269-t001:** Definitions of behaviors associated with acceleration events.

Behavior	Definition
Collision	Bird hits or moves into an object at moment of event. The object that the bird collided with is identified: high perch, low perch, floor, cage wall, or bird. The action that the bird performed immediately preceding the collision is also identified: ascent towards perch, descent from perch, slipping while walking on top of a perch, physical push from conspecific, or running into an object on the bird’s own.
Aggression	Bird fights with or confronts other birds, but no clear collision event is observed. Aggressive behavior typically involves pushing out chest or standing taller while orienting towards each other, flapping wings, chasing and jumping, and aggressive pecking directed at another bird, typically at the head.
Grooming	Bird is preening, cleaning, and aligning feathers using own beak, or ruffles feathers.
Wing flapping	Bird opens and extends both wings, but not due to collision event or aggressive interaction.
Mass scattering	All or large group of birds in cage move to one side of the cage (cage disturbance), with no clear collision observed for focal hen.
Cannot ID	Cannot identify acceleration event because hen behavior was obstructed from view, no clear behavior could be determined, or a video error occurred.

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
