# Peer review of "Pullet Rearing Affects Collisions and Perch Use in Enriched Colony Cage Layer Housing"

_animals, 2020, doi:10.3390/ani10081269_

Round 1
Reviewer 1 Report
Dear Author(s),
Congratulations with the presentation of this wonderful manuscript.
Your style of writing is very easy, well-structured and pleasant to read.
Please find included your manuscript in which I added my feedback.
Please pay extra attention to the use of 'successfulness' in how you calculated it (or could have calculated it) and how you define it.
Looking forward receiving the revised version.
Kind regards

Reviewer 2 Report
Manuscript Animals-856550, entitled “Pullet rearing affects collisions and perch use in enriched colony cage layer housing”
Recommendation: The above paper is not suitable for publication in its present form.
General comment
The article provides useful information about the effects of pullet rearing conditions on collisions and perch use in enriched colony cage layer housing. Although the experiment is appropriately designed and implemented, there are some points that should be corrected or clarified.
Specific comments
L19: “at the age of 23 weeks” instead of “to 23 weeks of age”
L20: “till the age of 73 weeks” instead of “out to 73 weeks of age”
L23: “…weeks of age, and were then removed into…”
L24: “…events in hens with tri-axial…”
L32: “…and were then removed into…”
L39: Please delete “specifically”
L39-40: “that offers increased opportunities of using” instead of “with opportunities to use”
L40-41: Please rephrase
L49: “improved” instead of “better”
L54: “…skeletal system remain till the end of the laying period. Aviary-reared…”
L85: “selected from” instead of “among”
L106: “…for the rest rearing period.”
L133-134: What do you mean by “Since hens were randomly selected within each ECC at each age, the same focal hens were not used for each age”? Were the responses of the other hens of the cage towards the hens with the jackets examined? The color of the jacket was white as the color of the hen hybrid? Could you provide a photo as a Supplementary material?
L147: “After the end” instead of “At the conclusion”
L150: “…assigned to videos from…”
L164: Please add a sentence that your data are presented as medians or means etc. Please specify for each case.
L196, 206, 215, 304: “Among” instead of “Of”
L202: “…AVI hens from the 12 ECC.”
L212-219: Why is aggression increased and is the most frequent cause of acceleration events at the age of 49-50 weeks old? Please provide an explanation in discussion.
L237: “…49 wks of age, respectively). Moreover…”
L240: “…49 wks of age, respectively).”
L256: “…decreased at 35 wks of age…”
L262: “…collisions/hen/10 d at 21, 35 and 49 wks of age, respectively). CON hens…”
L264: “…collisions/hen/10 d, respectively).”
L318: “…rearing conditions affects…”
L319-320: “…as perching behavior, in ECC housed laying hens.”
L320-323: Please delete (repetition)
L325: “with increased age” instead of “as they aged”
L326: “…hens at the initial and peak phase of the laying period (21 and 35 wks of age)…”
L335-336: “…enriched (AVI) rearing environment improves birds’ abilities to learn, which may…”
L347: “rate” instead of “incidence”
L348: “Therefore, by comparing our data with that of these…”
L355: “These authors” instead of “Baker et al.”
L359: “observed discrepancy” instead of “different numbers”
L363: “the majority of” instead of “most”
L368: “study” instead of “results”
L369: “with age” instead of “as birds aged”
L371-372: “the increased activity rates at an early age that are later reduced as the laying period continued” instead of “hens being the most active earlier in lay and reducing activity over time”
L377-378: “As a result, the increased acceleration events and collisions at the initial phase of the laying period observed in the present study could also be attributed to the novel environment.”
L380: “performed” instead of “doing”
L381: “constituted” instead of “made up”
L383: “are in agreement” instead of “correspond”
L390: “hyphotetized” instead of “predicted”. Please delete “overall”
L391: “finding” instead of “predicted effect”
L398-399: “…sufficient time to get accustomed with the use of perches. This is possible since the low perch…”
L400: “had” instead of “ran”
L405: Please delete “using the perches for”
L414: “other researchers” instead of “others”
L418: “discrepancies” instead of “differences”
L426: “prevented” instead of “dissuaded”
L432: “…previous studies found that the majority of collisions were displayed during…”
L433: “frequently” instead of “frequent”
L442-443: “…events, since that aggression was the second-most frequent cause of acceleration events a fact that was also shown in a previous study [17].”
